# A Novel Velocity-Based Control in a Sensor Space for Parallel Manipulators

**DOI:** 10.3390/s22197323

**Published:** 2022-09-27

**Authors:** Antonio Loredo, Mauro Maya, Alejandro González, Antonio Cardenas, Emilio Gonzalez-Galvan, Davide Piovesan

**Affiliations:** 1Facultad de Ingeniería, Universidad Autónoma de San Luis Potosí, San Luis Potosi 78290, Mexico; 2Tecnologico de Monterrey, Escuela de Ingenieria y Ciencias, Queretaro 76130, Mexico; 3Biomedical Engineering, Gannon University, Erie, PA 16541, USA

**Keywords:** vision-based control, camera-space manipulation, parallel robot

## Abstract

It is a challenging task to track objects moving along an unknown trajectory. Conventional model-based controllers require detailed knowledge of a robot’s kinematics and the target’s trajectory. Tracking precision heavily relies on kinematics to infer the trajectory. Control implementation in parallel robots is especially difficult due to their complex kinematics. Vision-based controllers are robust to uncertainties of a robot’s kinematic model since they can correct end-point trajectories as error estimates become available. Robustness is guaranteed by taking the vision sensor’s model into account when designing the control law. All camera space manipulation (CSM) models in the literature are position-based, where the mapping between the end effector position in the Cartesian space and sensor space is established. Such models are not appropriate for tracking moving targets because the relationship between the target and the end effector is a fixed point. The present work builds upon the literature by presenting a novel CSM velocity-based control that establishes a relationship between a movable trajectory and the end effector position. Its efficacy is shown on a Delta-type parallel robot. Three types of experiments were performed: (a) static tracking (average error of 1.09 mm); (b) constant speed linear trajectory tracking—speeds of 7, 9.5, and 12 cm/s—(tracking errors of 8.89, 11.76, and 18.65 mm, respectively); (c) freehand trajectory tracking (max tracking errors of 11.79 mm during motion and max static positioning errors of 1.44 mm once the object stopped). The resulting control cycle time was 48 ms. The results obtained show a reduction in the tracking errors for this robot with respect to previously published control strategies.

## 1. Introduction

Parallel robots are composed of mobile (end-effector) and fixed bases, which are linked together by at least two independent kinematic chains with *n* degrees of freedom [1]. By having more than one kinematic chain, parallel robots have some advantages over their serial counterparts, i.e., higher velocity and acceleration, improved rigidity, higher accuracy, better load-capacity-to-weight ratio, low inertia, good stability, and simple inverse kinematics. On the other hand, their main disadvantages are: reduced workspaces, complex kinematic models [2,3], limited workspaces, and multiple singular configurations in their reachable workspaces, resulting in limited maneuverability and low dexterity [4]. Traditional control strategies are hard to implement on parallel robots, due to: the robot’s multiple solutions to the direct kinematic problem and modeling errors in the Jacobian as a result of inaccurate geometric calibration. Research on parallel robots has focused on mitigating the effects of these drawbacks on their precision and ease of control [5,6,7,8].

Cameras are often used as effective robotic sensors, as they allow for non-contact exteroceptive sensing. They can help to compensate for errors in the robot’s kinematic model in positioning tasks as they provide measurements of the end-effector and the target. A control law based on these measurements can ensure precise positioning of the end-effector on the target despite uncertainties in the robot model. Visual Servoing (VS) and Camera Space Manipulation (CSM) are two of the most popular vision control approaches used in robotics. Vision-based control can be regarded as a sensor-based control strategy, which strongly resembles the process of our central nervous system [9].

VS control is used for real-time control, capable of integrating visual information obtained by one or more cameras in a closed-loop control cycle. Usually, VS algorithms are kinematic control schemes that do not include the stability analysis of closed-loop dynamics [10]. Its development allows for increased precision in manipulation tasks based on visual feedback [11,12,13]. VS can be classified based on the error signal it handles as: 3D (it is also known as “position-based” but this term is reserved in this work to describe control approaches where the control signal is the joint position), where the error is expressed in Cartesian coordinates, 2D (also called image-based, where the error is expressed in image coordinates), and some variants (mixing coordinates). With respect to the control signal it produces, VS can be classified as position-based in the sense that the control signal is the joint position (most 3D schemes are of this type) or velocity-based, in the sense that the control signal is the joint velocity (usually 2D schemes are of this type). Velocity-based controllers require the calculation of the robot Jacobian, and if using a 2D approach, it will also require what is known as the image Jacobian, i.e., the analytical transformation between the operational tangent space and the image tangent space. Image Jacobians depend on the visual features used in the approach and the camera parameters (a calibration process is required to obtain them) and can be complex and difficult to obtain [14]. Some work has also been done designing controllers robust to uncertainties in the Jacobian matrix as reported in [15].

On the other hand, CSM [16,17,18] and its variants, such as Linear Camera Model—Camera Space Manipulation (LCM-CSM) [19,20], are calibration-free and do not require the computation of the robot’s Jacobian since they are position-based. They only require the (uncalibrated) robot’s kinematic model and focus on controlling the end-effector position, refining its precision iteratively at each movement iteration. These approaches do not need the image Jacobian because they do not involve the velocities in the operational space or the image space since it is designed for static positioning of the robot using position commands. Previous attempts at tracking trajectories using vision-based control have used a dual-stage approach where velocity prediction was obtained in a first assessment stage, followed by a position control strategy trying to predict such velocity [21]. Tracking and movement using CSM require continuous updating of the view parameters, which can be performed non-linearly using a heuristic algorithm, such as a particle filter [22], or by local linearization conducive to the use of statistical estimators, such as the Kalman filter [18]. In [23], the authors presented a LCM-CSM variant where the tracking of moving objects was possible using positioning commands; however, no CSM velocity-based variant has been developed to this day. Velocity-based control could prove to be more energy efficient, suitable for low control frequencies, and robust to variations on the control signal [24,25].

Recent works in vision-based control have sought to enhance CSM with artificial intelligence (A.I.) techniques. The work reported in [26] combined the CSM method and a differential evolution technique to increase the precision of robot maneuvers; the authors of [27,28] used the CSM method and support the vector regression technique in robotics tasks to catch moving objects.

Vision-based systems with A.I. control strategies are very active research areas; examples are found in References [29,30,31]. In a recent study [32], the author presented a comprehensive survey on such topics. A drawback of these methods is the amount of data required to train the algorithm used for the tracking task. We opted for a more classical approach where the velocity of the target was estimated via a Kalman filter.

This work presents a new vision-based velocity control, inspired by LCM-CSM, meant to command the joint velocities of a robot to perform a task. This is done in a closed-loop control scheme by using the LCM-CSM vision parameters and the robot’s Jacobian. The contribution of the work presented herein involves the use of vision parameters to map the image space velocity vector to the end-effector velocity. This approach incorporates the benefits of velocity-based control while keeping some of the advantages of the traditional LCM-CSM, as presented in this paper.

This article is structured as follows: Section 2 presents a review of the kinematic model of Delta robots. Section 3 introduces the traditional CSM and LCM-CSM techniques while Section 4 describes the proposed modifications necessary to enable velocity control. Section 5 describes the experimental platform, and Section 5.4 details the experiments performed. Section 6 presents the results, and offers a brief discussion of them. Finally, Section 7 presents our conclusions and shows possible venues for future work.

## 2. Delta-Type Parallel Robots

Figure 1 shows the PARALLIX LKF-2040, an academic Delta-type robot composed of a mobile platform attached to a fixed base by three closed kinematic chains. Each chain has a parallelogram guaranteeing parallelism between the platforms [33]. PARALLIX LKF-2040 is capable of imparting three translational degrees of freedom to its end-effector. Figure 1 also shows the robot’s geometric parameters required for its kinematic model and is described in [34]. These are: the radii of the fixed (*R*) and mobile platform (*r*), the lengths of the actuated links ($L_{1}$), the parallelogram ($L_{2}$), the constant angle of each arm with respect to the robot’s coordinate system ($\alpha_{i}$), the displacement angles of the active joints ($\theta_{i1}$), and the passive joints ($\theta_{i2}$ and $\theta_{i3}$).

### 2.1. Forward and Inverse Kinematic Model of the Delta Robot

The aforementioned geometric parameters define the configuration of each kinematic chain. With this information, it is possible to compute the robot’s implicit forward kinematic model shown in (Equation 1) as well as its inverse kinematic model given by (Equation 3) [35] as follows:(1)$${\left(X_{i}-X_{p}\right)}^{2}+{\left(Y_{i}-Y_{p}\right)}^{2}+{\left(Z_{i}-Z_{p}\right)}^{2}={L_{2}}^{2}$$
where:(2)$$\begin{array}{cc} \; & X_{i}=\left(R+L_{1}\cos\theta_{i1}-r\right)\cos\alpha_{i}\\ \; & Y_{i}=\left(R+L_{1}\cos\theta_{i1}-r\right)\sin\alpha_{i}\\ \; & Z_{i}=-L_{1}\sin\theta_{i1} \end{array}$$
and $\left(X_{p},Y_{p},Z_{p}\right)$ denote the position of the center of the mobile platform expressed with respect to a frame attached to the center of the fixed platform.
(3)$$\tan\frac{\theta_{i1}}{2}=\frac{-2Z_{p}\pm \sqrt{4{Z_{p}}^{2}+4{R_{1}}^{2}-S^{2}+{Q_{i}}^{2}\left(1-\frac{{R_{1}}^{2}}{{L_{1}}^{2}}\right)+Q_{i}\left(\frac{-2R_{1}S}{L_{1}}-R_{1}\right)}}{-2R_{1}-Q_{i}\left(\frac{R_{1}}{L_{1}}-1\right)-S}$$
with:(4)$$\begin{array}{cc} \; & Q_{i}=2X_{p}\cos\alpha_{i}+2Y_{p}\sin\alpha_{i}\\ \; & R_{1}=R-r\\ \; & S=\frac{1}{L_{1}}\left(-{X_{p}}^{2}-{Y_{p}}^{2}-{Z_{p}}^{2}+L_{2}^{2}-L_{1}^{2}-{R_{1}}^{2}\right) \end{array}$$

### 2.2. Computation of the Delta Robot’s Jacobian

The Jacobian matrix provides a transformation between the end effector velocity vector ($\dot{x}$) in the Cartesian space, and the joint velocity vector ($\dot{q}$). The Jacobian matrix of the Delta robot is given by [36] as:(5)$$\begin{array}{c} \dot{q}=J_{q}^{-1}J_{x}\dot{x} \end{array}$$
where $J_{x}$ and $J_{q}$ are the [3×3] matrices obtained from the derivation of the kinematic model. Their elements are given by:(6)$$\begin{array}{c} J_{x}=\left[\begin{array}{ccc} a_{11} & a_{12} & a_{13}\\ a_{21} & a_{22} & a_{23}\\ a_{31} & a_{32} & a_{33} \end{array}\right] \end{array}$$
and
(7)$$\begin{array}{c} J_{q}=diag\left(b_{11},b_{22},b_{33}\right) \end{array}$$
where:(8)$$\begin{array}{cc} \; & a_{i1}=\sin\left(\theta_{i3}\right)\cos\left(\theta_{i1}+\theta_{i2}\right)\cos\left(\alpha_{i}\right)+\cos\left(\theta_{i3}\right)\sin\left(\alpha_{i}\right)\\ \; & a_{i2}=\sin\left(\theta_{i3}\right)\cos\left(\theta_{i1}+\theta_{i2}\right)\sin\left(\alpha_{i}\right)-\cos\left(\theta_{i3}\right)\cos\left(\alpha_{i}\right)\\ \; & a_{i3}=-\sin\left(\theta_{i3}\right)\sin\left(\theta_{i1}+\theta_{i2}\right)\\ \; & b_{ii}=L_{1}\sin\left(\theta_{i2}\right)\sin\left(\theta_{i3}\right) \end{array}$$

It should be mentioned that both $J_{x}$ and $J_{q}$ are dependent on the robot’s current configurations ($b_{ii}$, $a_{i1}$, $a_{i2}$, and $a_{i3}$). That is, their numerical values depend on the instantaneous end-effector positions.

## 3. Camera-Space Manipulation with a Linear Camera Model

The LCM-CSM technique works by mapping visual markers from the operational space to the camera space. In an initial, offline stage, markers positioned in the robot’s end effector are used. The corresponding three-dimensional positions of the markers are obtained by using the robot’s forward kinematic model and, together with their correspondences in the camera space, are used to estimate the parameters of the mapping. If the robot has to perform a positioning task, such a mapping has to be determined for each of at least two participating cameras [19].

The LCM-CSM technique is a variant of the classic CSM method, which replaces the orthographic camera model with the pin-hole model [19]. LCM-CSM defines the so-called ‘*vision parameters*’ by solving a set of linear equations.

In general, a relationship between points in three-dimensional spaces and their planar projections [37] in sensor spaces, can be written as,
(9)$$\begin{array}{c} \rho \left[\begin{array}{c} u_{i}\\ v_{i}\\ 1 \end{array}\right]=\left[\begin{array}{cccc} \begin{array}{c} p_{11}\\ p_{21}\\ p_{31} \end{array} & \begin{array}{c} p_{12}\\ p_{22}\\ p_{32} \end{array} & \begin{array}{c} p_{13}\\ p_{23}\\ p_{33} \end{array} & \begin{array}{c} p_{14}\\ p_{24}\\ 1 \end{array} \end{array}\right]\left[\begin{array}{c} \begin{array}{c} x_{i}\\ y_{i} \end{array}\\ \begin{array}{c} z_{i}\\ 1 \end{array} \end{array}\right] \end{array}$$
where the center of the *i*th visual marker, located on the end-effector, is represented by the coordinates $\left(u_{i},v_{i}\right)$ expressed in pixels, while ρ is a scaling factor, which, under the model’s assumptions, tends to a value of one. Cameras are placed far enough to neglect the effect of the camera perspective. For the work presented herein, the robot’s workspace was small enough that it could be mostly sampled during the execution of a pre-planned trajectory (also known as “pre-plan”) in which samples of the marker’s three-dimensional locations and the corresponding camera-space locations were obtained. A requirement of such a trajectory is that samples encompass a large region of the manipulator’s workspace. The matrix $\left[\begin{array}{cccccc} p_{11} & p_{12} & p_{13} & ... & p_{33} & 1 \end{array}\right]$ encompasses the vision parameters and the vector ${\left[\begin{array}{ccc} x_{i} & y_{i} & z_{i} \end{array}\right]}^{T}$ describes the three-dimensional position of the *i*th visual marker, with respect to the global reference frame.

A linear system for the determination of the camera parameters was obtained from the previous expression, as follows:(10)$$Y_{i}=A_{i}\;P$$
where
(11)$$Y_{i}={\left[\begin{array}{cc} u_{i} & v_{i} \end{array}\right]}^{T}$$
(12)$$A_{i}=\left[\begin{array}{ccccccccccc} x_{i} & y_{i} & z_{i} & 1 & 0 & 0 & 0 & 0 & -u_{i}x_{i} & -u_{i}y_{i} & -u_{i}z_{i}\\ 0 & 0 & 0 & 0 & x_{i} & y_{i} & z_{i} & 1 & -v_{i}x_{i} & -v_{i}y_{i} & -v_{i}z_{i} \end{array}\right]$$
(13)$$P={\left[\begin{array}{ccccc} p_{11} & p_{12} & \cdots & p_{32} & p_{33} \end{array}\right]}^{T}$$

In order to obtain vector P, at least 6 linearly-independent samples are required (i=1,2,…,6). If more than 6 points are used in the computation of P, an over-constrained system is obtained and a minimization process is required. The optimal value of P can be computed using least squares optimization:(14)$$\begin{array}{c} \hat{P}={\left(A^{T}A\right)}^{-1}A^{T}Y \end{array}$$
where $Y={\left[u_{1}\;v_{1}\;\cdots \;u_{n}\;v_{n}\right]}^{T}$ are the measured or observed values, $A={\left[{A_{1}}^{T}\cdots {A_{n}}^{T}\right]}^{T}$ and n≥6 is the number of points used. Typically, the view parameters are first (roughly) estimated using the set of (global) samples from the “pre-plan”. It is immediate to see that the regression between the measurements in the camera space and the estimate of the markers in the operational space allows bypassing the need for a precise image Jacobian, which is *de facto* incorporated in the uncertainty of the view parameters. Then, as the robot approaches the (static) target, new (local) samples are obtained in the neighborhood of the final position and are incorporated in (Equation 14) to refine the parameters compensating for the uncertainties of the models. Usually, these local samples are given more significance by assigning them a weight (scalar factor) greater than one; each new sample has a higher weight.

### Varying Weights

A problem that needs to be addressed, to enable an accurate estimation of the target point, is the updating of the view parameters when the target is in motion, as reported in [23]. Indeed, when the target is moving, the new samples are dispersed along the trajectory rather than accumulating near the target and are not as effective to refine the parameters.

In order to solve this problem, in the proposed approach, the set of samples obtained in the “pre-plan” was used throughout the experiment. A relative weight was applied to each sample and larger weights were given to samples located closer to the moving target in each image frame. Note that no new samples were required to refine the view parameters, thus the corresponding image acquisition and processing time were saved, yielding a lower computational cost compared to the usual approach. A drawback of this approach involves guaranteeing a sufficiently high number of pre-plan samples to cover the workspace appropriately.

The weight given to each sample can be defined using a diagonal matrix W as,
(15)$$\begin{array}{c} W=\left[\begin{array}{ccc} w_{1} & 0 & 0\\ 0 & \ddots & 0\\ 0 & 0 & w_{m} \end{array}\right] \end{array}$$
where $w_{i}$, refers to the *i*th “pre-plan” sample. If W is used to estimate P, the following equation can be used for the estimation of the optimal parameters, with different weights associated with each sample,
(16)$$\begin{array}{c} \hat{P}={\left(A^{T}\mathrm{WA}\right)}^{-1}A^{T}WY \end{array}$$

In the work presented herein, the following equation is proposed for the weight given to samples used to determine the camera parameters,
(17)$$w_{i}=\frac{k}{l_{i}}$$
where, according to [38], such a definition enables a consistent positioning error. The parameter *k* is defined for the particular maneuver while $l_{i}$ represents the Euclidean distance between each sample acquired during the pre-planned trajectory, and the location of the maneuvering objective.

## 4. CSM-Based Velocity Control

### Control Law

Consider (Equation 9) for a point attached to the robot’s end-effector; by taking its time derivative, a mapping relating the three-dimensional velocity (mm/s) of the end-effector and its associated velocity in the image plane (pixels/s) was obtained:(18)$$\begin{array}{c} \left[\begin{array}{c} \dot{u}\\ \dot{v} \end{array}\right]=\left[\begin{array}{c} \begin{array}{ccc} p_{11} & p_{12} & p_{13} \end{array}\\ \begin{array}{ccc} p_{21} & p_{22} & p_{23} \end{array} \end{array}\right]\left[\begin{array}{c} \dot{x}\\ \dot{y}\\ \dot{z} \end{array}\right] \end{array}$$

Note that mapping (Equation 18) is not injective. By adding at least another camera, the injective mapping (Equation 19) could be established.
(19)$$\begin{array}{c} \left[\begin{array}{c} {\dot{u}}^{1}\\ {\dot{v}}^{1}\\ {\dot{u}}^{2}\\ {\dot{v}}^{2} \end{array}\right]=\left[\begin{array}{ccc} p_{11}^{1} & p_{12}^{1} & p_{13}^{1}\\ p_{21}^{1} & p_{22}^{1} & p_{23}^{1}\\ p_{11}^{2} & p_{12}^{2} & p_{13}^{2}\\ p_{21}^{2} & p_{22}^{2} & p_{23}^{2} \end{array}\right]\left[\begin{array}{c} \dot{x}\\ \dot{y}\\ \dot{z} \end{array}\right] \end{array}$$
where the superindex identifies the corresponding camera. This procedure can be expanded to include any additional number of cameras, but for the sake of simplicity, the deduction was performed using only two cameras. Equation (Equation 19) can be written in a compact form,
(20)$$\begin{array}{c} \dot{s}=J_{csm}\dot{x} \end{array}$$
where $\dot{s}$ is the time derivative of $s={\left[\begin{array}{cccc} u^{1} & v^{1} & u^{2} & v^{2} \end{array}\right]}^{T}$, $\dot{x}$ is the time derivative of $x={\left[\begin{array}{ccc} x & y & z \end{array}\right]}^{T}$, and $J_{csm}$ is the *velocity CSM matrix*, defined as:(21)$$J_{csm}=\left[\begin{array}{ccc} p_{11}^{1} & p_{12}^{1} & p_{13}^{1}\\ p_{21}^{1} & p_{22}^{1} & p_{23}^{1}\\ p_{11}^{2} & p_{12}^{2} & p_{13}^{2}\\ p_{21}^{2} & p_{22}^{2} & p_{23}^{2} \end{array}\right]$$

For a given positioning task, the error signal in the camera space is given by:(22)$$\begin{array}{c} e=s^{*}-s \end{array}$$
where $s^{*}={\left[\begin{array}{cccc} u^{1*} & v^{1*} & u^{2*} & v^{2*} \end{array}\right]}^{T}$ is the target point and s (as defined before) corresponds to a point attached to the end-effector. Both points are expressed by their pixel coordinates on each participating camera.

For a positioning task, by taking the time derivative of (Equation 22) and considering the velocity CSM matrix (Equation 21) the evolution of the error can be written as:(23)$$\dot{e}={\dot{s}}^{*}-J_{csm}\dot{x}$$

By using (Equation 5), the previous equation can be rewritten as:(24)$$\dot{e}={\dot{s}}^{*}-J_{csm}J_{x}^{-1}J_{q}\dot{q}$$

The closed-loop behavior of the error is proposed ideally as $\dot{e}=-Ke$ where K is a control gain matrix. Note that the system is such that if K is defined as a positive definite matrix, the error vector will exponentially decrease to zero. By substitution in (Equation 24), we find that:(25)$$-Ke={\dot{s}}^{*}-J_{csm}J_{x}^{-1}J_{q}\dot{q}$$

Solving for $\dot{q}$ and using (Equation 22), the following control law is synthesized:(26)$$\dot{q}=J_{q}^{-1}J_{x}J_{csm}^{†}\left[K\left(s^{*}-s\right)+{\dot{s}}^{*}\right]$$
where $J_{csm}^{†}$ is the Moore–Penrose pseudoinverse of (Equation 21) (alternatively, the last row of $J_{csm}$ can be eliminated, since it provides redundant information in the image space, yielding a square, invertible matrix and $J_{csm}^{-1}$ can be used instead). In (Equation 26), $s^{*}$ is obtained by placing a marker in the target object and measuring its projection in the camera space, K is user defined, ${\dot{s}}^{*}$ is not known in the general case and cannot be directly measured (but could be estimated from measurements), and the rest of the elements are known as described before. Figure 2 shows the block diagram of the closed-loop control law.

While tracking moving objects, if ${\dot{s}}^{*}$ is not compensated (i.e., it is set to zero in the control law), tracking errors manifest as drag errors where the robot’s end-effector follows the moving object at a certain distance. However, as mentioned before, it is possible to obtain an estimate of the object’s camera space velocity through other means, for example, a Kalman filter (KF) as in [23]. The present work does not focus on this estimation; therefore, the implementation of [23] with previously tuned covariance matrices is used for this purpose in this control approach. The key elements of this Kalman filter targeting velocity estimation in camera space are presented in the next subsection, more details can be found in the above reference.

Please refer to Appendix A for more details regarding the computation of a Kalman filter.

## 5. Materials and Methods

### 5.1. Hardware

The developed control law was implemented on the PARALLIX LKF-2040, a 3 DOF Delta-type parallel robot (Figure 3), equipped with a variable speed conveyor belt as shown in Figure 4. A vision system consisting of two uEye UI-2210SE-M-GL USB digital cameras with a resolution of 640 × 480 pixels was used to provide the images for the controller. The cameras were placed 1.5 m away from the center of the robot’s workspace and at a distance of 0.9 m from each other. Figure 3 shows the positions of the cameras with respect to the Delta PARALLIX LF-2040. The controller was implemented using an Intel Core i5-6600HQ, 2.5 GHz, 64-bit processor and 8 GB RAM computer running Windows 10.

### 5.2. Software

The control system was developed in C++ using Visual Studio 2012 Express. Additionally, matrix operations implemented were conducted using the GNU-GSL libraries [39]; image acquisition and processing were performed using OpenCV [40]. Finally, the control system was implemented using multiple threads in order to decrease its cycle time. The main thread was devoted to the control law calculations, while secondary threads were used for image acquisition and processing. The resulting control cycle time was 48 ms.

### 5.3. Position Measuring System

In order to measure the 3D tracking error during the tasks, the following was used.

For static trajectories, a vernier caliper was used to measure the 3D positioning error.

For moving trajectories, a vision-based measuring system (VMS) was implemented. The measuring system relies on the varying weights LCM-CSM algorithm (Equation 9)–(Equation 17). Two participating cameras (Equation 9) can be written as
(27)$$\begin{array}{c} \left[\begin{array}{c} u^{1}\\ v^{1}\\ u^{2}\\ v^{2} \end{array}\right]=\left[\begin{array}{ccc} p_{11}^{1} & p_{12}^{1} & p_{13}^{1}\\ p_{21}^{1} & p_{22}^{1} & p_{23}^{1}\\ p_{11}^{2} & p_{12}^{2} & p_{13}^{2}\\ p_{21}^{2} & p_{22}^{2} & p_{23}^{2} \end{array}\right]\left[\begin{array}{c} x\\ y\\ z \end{array}\right] \end{array}$$

Once the corresponding vision parameters are obtained, it is possible to recover the 3D position of an observed image feature point.

The measuring system was calibrated using two visual markers placed on the end effector and separated 25 mm from each other. The markers were moved along a “pre-plan” trajectory sampling the robot’s workspace. On each selected position along the pre-plan, the 3D position of each visual marker was obtained as well as the distance between them. An average error of 0.37833 mm with a standard deviation of 0.26058 mm was obtained for this vision-based measuring system. It is noteworthy that the precision of this system can be improved by using higher resolution images and/or more cameras. On the other hand, the system can be used to measure the tracking error while the target moves along an arbitrary trajectory.

A root mean squared error was obtained with the instantaneous measurements on each cycle of the system. Only the second half of the total iterations in each task were used to calculate this error as it was intended to obtain the steady-state error of the task.

### 5.4. System Operation

The cameras’ vision parameters were estimated (see Section 3) before performing the positioning tasks. A total of 410 visual markers were used to initialize the vision parameters defined in (Equation 10), and their known three-dimensional and camera space positions were stored in memory. The markers were distributed evenly along the robot’s workspace, taking care to span the available volume. In this way, the vision parameter could be determined using a weight-based scheme as a function of the position of the point of interest. This improved the performance of the LCM-CSM method. Under this scheme, the visual marker positions, which were closer to the point of interest, were deemed more important and, thus, carried a larger weight during the estimation of the vision parameters [38]. That is, for the results shown here, the vision parameters were estimated anew each time the point of interest changed position either in three-dimensional or the camera space.

The proposed control scheme was tested using a series of positioning tasks. Three different sets of experiments were performed:(1)The first set of experiments consisted of a series of static positioning tasks, i.e., using the proposed control, the robot tracked a static target placed randomly inside the robot’s workspace. Ten tasks were executed, each repeated 3 times, for a total of 30 trials. Once the robot’s end-effector was within 2 mm of the target position, the robot held its position for 50 control cycles, and the task’s mean squared error was computed. During these experiments, the control gain matrix (K) was chosen of the form K=kI where *I* is the 3×3 identity matrix and k=2.3; this value was tuned heuristically.(2)For the second set of experiments, the target was placed on a conveyor belt (approximately in the middle of the belt, width-wise) running at a constant speed. That is, the target moved following a linear trajectory referred to as “constant speed linear trajectory”. Three different speeds were used: 7, 9.5, and 12 cm/s.Additionally, three different control gain matrices (K) were tested, of the form K=kI, where k=2.3,2.7,3.1. The value of *k* was chosen as large as possible while maintaining no osculations on the task’s positioning response.For each speed, the task was performed 10 times with each of the possible K matrices. Finally, each task was also carried out under two conditions regarding the target’s velocity (${\dot{s}}^{*}$) compensation in the control law; (1) an estimate obtained by means of a Kalman filter was used, and (2) no estimation was used (the compensation was set to zero) yielding a simpler implementation but producing a larger error. However, this error can be reduced by increasing the control gain. In each case, the robot’s tracking error was measured.(3)For the last set of experiments, the target was moved freehand along different trajectories inside the robot’s workspace. This experiment was referred to as “freehanded trajectory”. These trajectories were: a circle, a square, an eight shape, a lemniscate, a zig-zag, and a decreasing spiral. The control gain matrix (K) was chosen to be a diagonal matrix with a value of 2.7 on its non-zero terms.

For each positioning task, the following steps were developed:(1)We obtained the coordinates of the objective point in pixels (s*) and loaded the vision parameters previously estimated during the “pre-plan”.(2)The program entered the control cycle, setting a convergence criterion of error in each camera coordinate ($u_{i},v_{i}$) less than 1 pixel.(3)The centroid of the visual marker attached to the robot’s end-effector ($s^{*}$) was obtained and the error was computed (Equation 22).(4)We read the robot’s encoders and found $\left[\begin{array}{ccc} \theta_{i1} & \theta_{i2} & \theta_{i3} \end{array}\right]$ to later calculate the Jacobian, as shown in (Equation 6) and (Equation 7).(5)We performed (Equation 19) to obtain the speeds to be injected into the robot’s controller.(6)We repeated until the convergence was obtained.

Figure 5 shows a flow diagram illustrating the previous steps.

## 6. Results and Discussion

### 6.1. Results

The LCM-CSM view parameters were initially estimated for the robot’s workspace. A total of 410 evenly distributed point positions in the Delta PARALLIX workspace had dimensions of 250×250×225 mm (l × w × h). The workspace was sampled using a grid with a 50 by 50 mm separation between the adjacent points on the x−y plane, and 25 mm on the *z*-axis.

The proposed control law (based only on the observations of the camera space and the robot’s joint space) was applied to the robot, and its performance was then evaluated in three different tasks.

For a set of 30 trials of *static positioning tasks*, an average positioning error of 1.086 mm was found (see Table 1). Figure 6a shows the decreasing error signal, measured in pixels, in the camera space. This was observed as an approach of the end-effector to the target position, visible in the same figure. Figure 6b shows the joint velocity vector. Note that the robot’s end-effector stopped as it reached close to the target.

For the *constant speed, linear trajectory tracking tasks*, a tracking or dragging error ranging from 9 to 28 mm was found while running 30 experiments of this type. This error seemed to be a function of the target/conveyor speed and the control gain (K). These experiments, and their associated dragging errors, are summarized in Table 2. Figure 7 shows a typical example of the position error’s evolution of error in both camera space and Cartesian space. This figure presents results at different target speeds. It is worth noting that the dragging error increases as a function of the target speed.

Finally, tracking results for the *freehand trajectories* are shown in Table 3. In this case, the robot’s end-effector followed the target before it stopped at a point inside the robot’s workspace. The positioning accuracy for this final point is shown in Table 3 labeled as the *final error*. A RMS tracking error on the order of 10 mm during the motion of the target and a final positioning error on the order of 1.4 mm were observed for these experiments. Figure 8 shows the freehand trajectories as well as the position errors in both the camera and the Cartesian spaces.

### 6.2. Discussion

From the results obtained for the *static positioning tasks*, on a set of 30 positioning maneuvers, the precision achieved by the implemented control was on the order of 1 mm. This is consistent with existent implementations of CSM-based position control laws, which are sufficiently accurate for many industrial tasks [20]. Finally, it is worth noting that the parallel robot used in this work was an academic prototype with a precision of 3.17 mm for static positioning when its original kinematic controls were used [23]. Moreover, the original kinematic controls did not allow for the tracking of a-priori unknown trajectories. Based on this, we can say that the proposed control scheme improves the performance of the platform.

For the experiments involving *constant speed linear target trajectories*, there were relatively high tracking/dragging errors throughout the experiment. This was the result of the object not stopping during the maneuver and was likely a consequence of an incorrect estimation of the target speed. Several factors may contribute to this, above all, the constant control cycle time. As the target’s speed increased, the convergence of the Kalman filter degraded. This is because the distance traveled by the target was larger between samples while the sampling rate remained constant. Another factor was the small but rapid variations of the conveyor belt’s speed. This might be due to the electronics in the speed variator driving the conveyor’s motor at a very low speed. These issues were not particular to the proposed control scheme. In any controlled system, the efficiency of the control scheme implementation was dependent on the hardware as well as on the relative speed of the controller dynamics and the dynamics of the physical systems. From the results summarized in Table 2, it can be observed that the tracking error increased as the target’s speed increased. It can also be seen that the tracking error decreased as the control gain increased. This was to be expected as the control gain acts directly on the positioning error. The obtained error can still be manageable for a number of industrial applications and it can be further reduced by improving the speed estimation using a higher rate sampling (provided the use of higher performance hardware) and finer tuning of the control gain.

For the case of the *freehand trajectories*, there was no control applied on the target trajectory and, therefore, they could be considered similar to arbitrary trajectories. The tracking errors for these types of trajectories are of the same order as those of linear trajectories, even when there is no knowledge of the target speed (the target motion was performed at speeds close to 7 cm/s). The final positioning precision was consistent with that achieved during the static positioning tasks (c.f. Table 1). These errors could be reduced by decreasing the image acquisition times, which, in this implementation, were hardware-dependent.

These results corroborate the ability of the proposed velocity-based CSM variant to track moving objects, something not achievable by most traditional CSM approaches or the PARALLIX’s original kinematic control scheme. Therefore, the proposed control scheme is an improvement over existing CSM approaches and the original control of the robot. For static tasks, the proposed approach achieves a positioning error of the same order of magnitude as other existing CSM approaches; this error is smaller than the one reported for the original robot non-CSM control [23].

In Figure 9 and Table 4 is shown a comparison of this work with significant works reported in the literature regarding vision-based robot controllers for different robotic architectures. In the figure, the coordinates of the center of each ball are given by the cycle time (in ms) and the average positioning error (mm); the radius of each ball is the standard deviation (mm) of the positioning error. It can be seen that this work produces a low positioning error with small standard deviation and an good cycle time. Table 4 shows the comparison in numbers as well as tracking errors for moving targets and provides the references to those works.

We also compared our work with some recent work in the literature that utilized both vision-based and non-vision-based controllers for Delta-type parallel robots.

The work in [3] focused on the tracking error of a 3DOF parallel manipulator using a visual control. Figures 9–11 in [3] report the error and displacement on each axis were plotted. The graphs show that the ratio between the maximum error and the maximum displacement within the workspace was about 0.01.

In [45], the tracking error of a Delta robot was controlled using computer vision. In this case, the error was calculated as the difference between the actual and desired angle of the joint. The error was presented as an angle where the peak error at the joint was 0.025 rad. Given that the trajectory in the Cartesian space was a circle with a diameter of 0.1 m we estimated that the rotation of the motor could not be larger than 1 rad given the dimension of the robot. Hence, the ratio of error to workspace was about 0.025.

In [8], the same type of Delta robot was controlled with a non-linear control. It is easy to see from Figures 8 and 9 in [3] at the maximum error of the end effector position was 0.006 m for a maximum displacement of 0.15 m, for a ratio of 0.04.

Finally, in [41], the error of a parallel robot in the 3D visual tracking of a ball, at velocities up to 1 m/s, controlled with a visual system, is presented. The oscillation of the ball in each direction was about 200 mm, and the tracking error was up to 15 mm, producing a ratio of 0.075

In our work, the tracking error varied from 8.89 mm at 7 cm/s to a maximum of 18.65 mm at 12 cm/s. Considering that each axis of the Delta robot in question was 250 mm, the ratio of our controller was capable of varying between 0.035 and 0.075. It can be seen that depending on the speed of our controller, it is comparable to other controllers presented in the literature. We demonstrated that CSM controllers, which are usually utilized for point-to-point pick-and-place, could also be used in tracking tasks, using the modification we proposed to the controlling algorithm.

## 7. Conclusions

This article presents the development and implementation of a novel velocity-based control on Camera Space Manipulation (CSM) applied to a Delta-type parallel robot. While CSM is often implemented as a position-based control, the proposed new methodology uses velocity-based control, which is the main contribution of this work. This contrasts with previous work, such as [23], where the control signal was defined in the position only. The proposed control has the ability to manipulate static and moving targets, even when the trajectory of the target is not known in advance, in contrast, CSM was originally designed to manipulate only static targets.

It should be noted that image-based Visual Servoing (VS) control techniques share some characteristics with the proposed velocity-based CSM technique. However, in VS, the image Jacobian has to be recalculated in each control cycle because it depends on the position of the target point within the image plane and the corresponding location along the focal axis of the camera. In contrast, in the case presented herein, the Jacobian matrix associated with the vision scheme ($J_{csm}$) can be regarded as a constant if the positioning task is performed inside the volume that contains the pre-plan trajectory.

Using the proposed control strategy, in the case of static objectives, the average positioning error was 1.086 mm with a standard deviation of 0.195 mm and an average task duration of 1.79 s. These results are competitive when compared against a number of industrial applications. Regarding mobile object tracking tasks, the results show tracking errors of 8.89, 11.76, and 18.65 mm for targets moving along the linear trajectory at speeds of 7, 9.5, and 12 cm/s, respectively; and 11.79 mm in the case of an arbitrary freehand trajectory during motion, with a maximum final static positioning error of 1.44 mm once the object stopped. Although the tracking error is relatively large while the target is moving, when the target stops, the proposed control can achieve similar precisions for static positioning tasks.

Traditional control schemes are hardly applicable to Delta robots and other parallel robots in general, due to open problems in the literature, such as the determination of the direct kinematic and dynamic models and singularity analyses, among others. Commonly, Delta robots are used in industrial applications for “pick-and-place” tasks; however, these applications require online adjustments due to the handling of different products on the same line, specifications by the client, and the displacements of the moving targets. Moreover, vision control techniques replace traditional robot control schemes; the authors believe that, in these cases, the proposed vision-based control via CSM can provide a suitable solution.

## Figures and Tables

**Figure 1 sensors-22-07323-f001:**
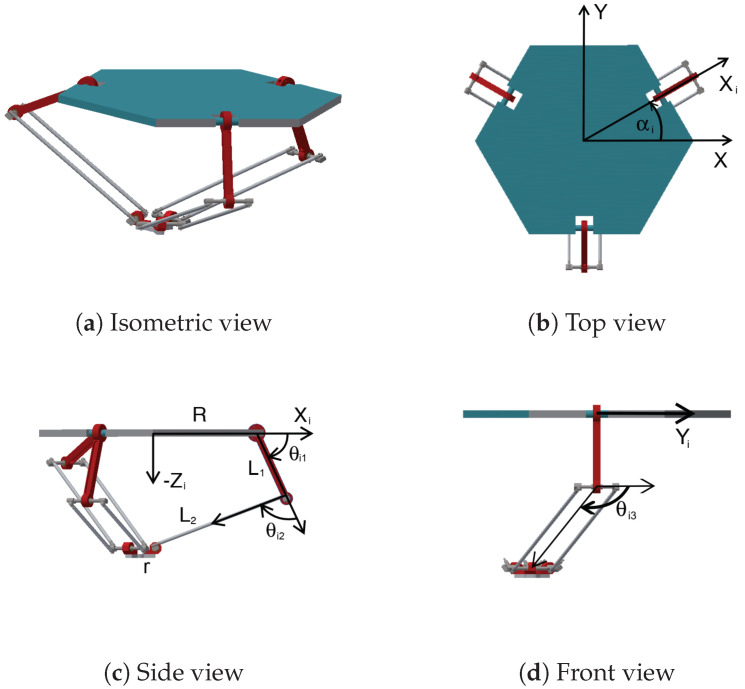
The PARALLIX LKF-2040, an academic Delta-type robot.

**Figure 2 sensors-22-07323-f002:**
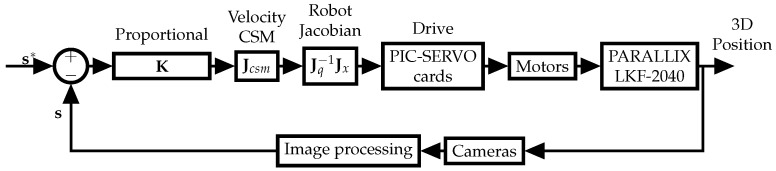
Block diagram of the proposed control scheme.

**Figure 3 sensors-22-07323-f003:**
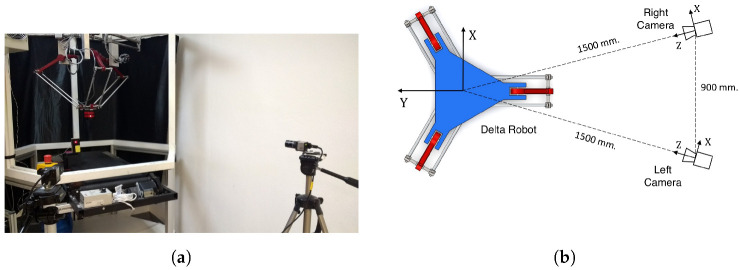
Experimental setup. (**a**) Delta PARALLIX LF-2040 and camera setup. (**b**) Delta robot and camera position scheme.

**Figure 4 sensors-22-07323-f004:**
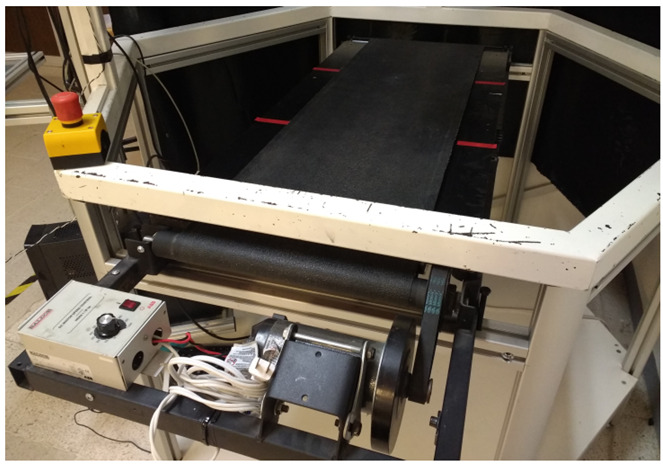
Conveyor belt.

**Figure 5 sensors-22-07323-f005:**
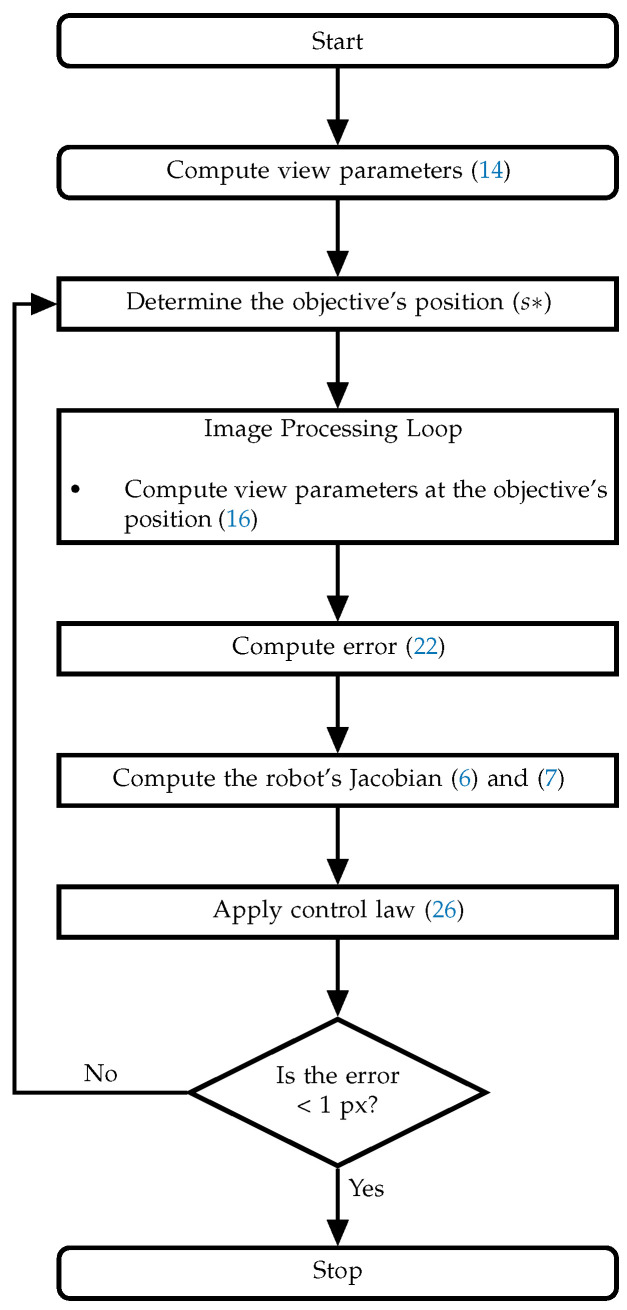
Control algorithm.

**Figure 6 sensors-22-07323-f006:**
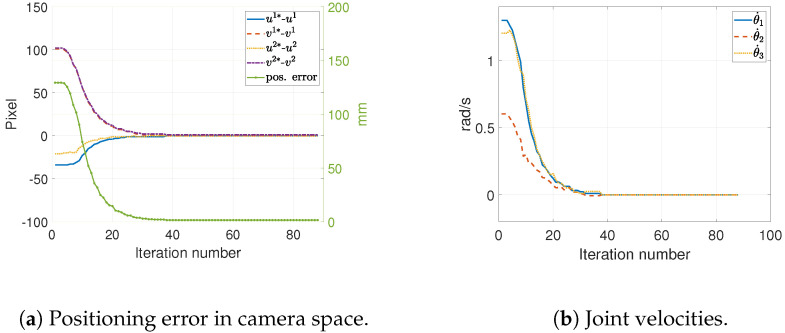
Error in camera space and joint velocities during a typical static positioning task.

**Figure 7 sensors-22-07323-f007:**
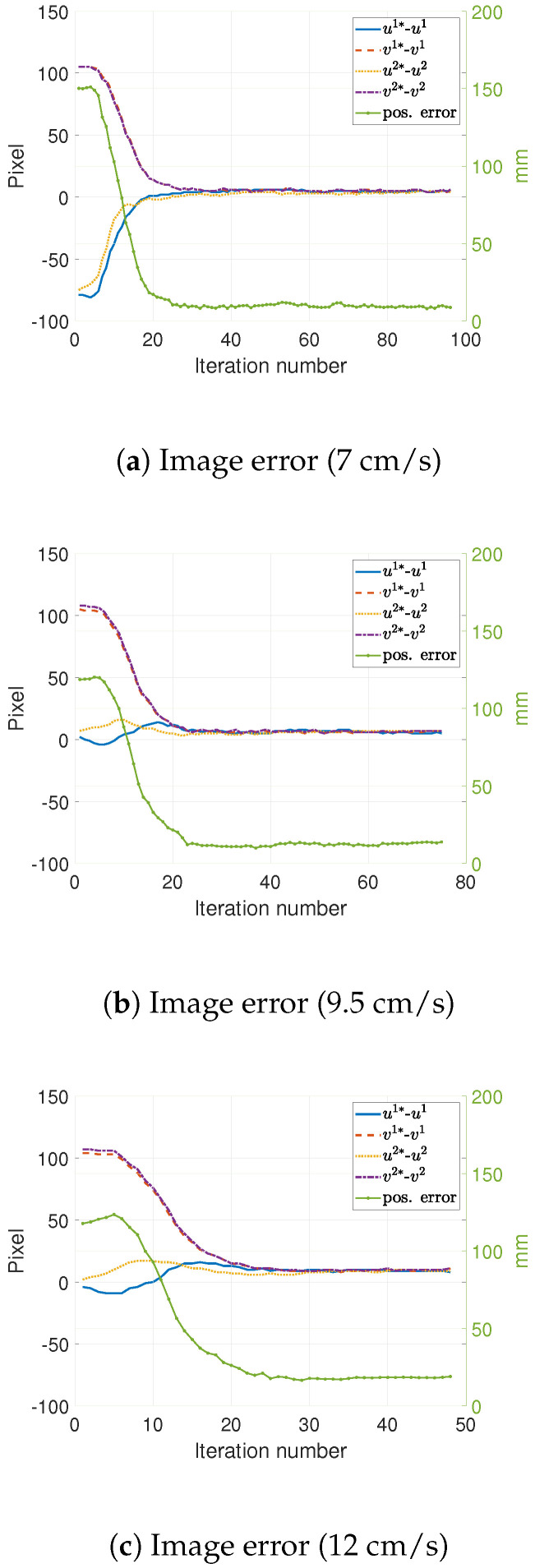
Tracking errors in 2D and 3D for a target in linear motion at a constant speed (the number in the parenthesis indicates the target speed).

**Figure 8 sensors-22-07323-f008:**
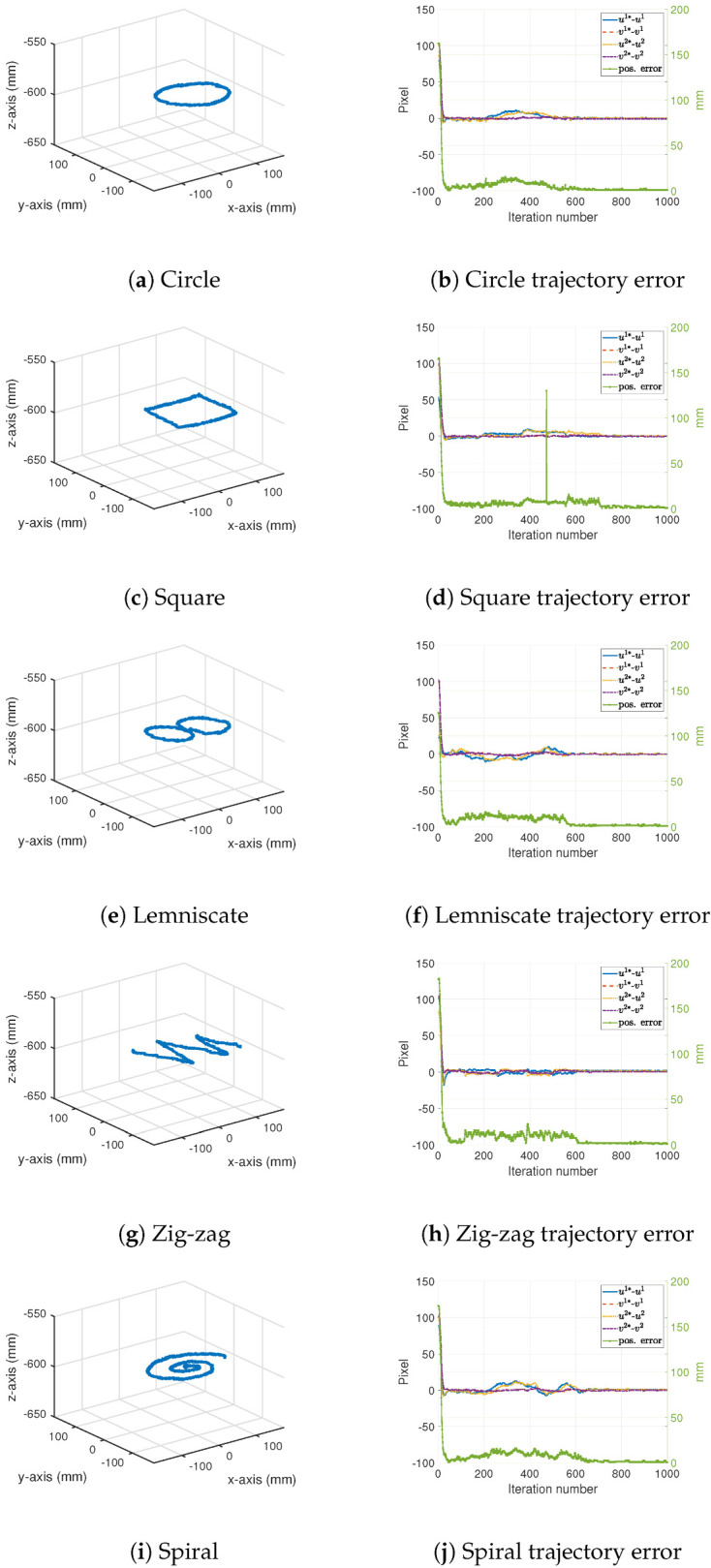
Measured target trajectories during freehand motion.

**Figure 9 sensors-22-07323-f009:**
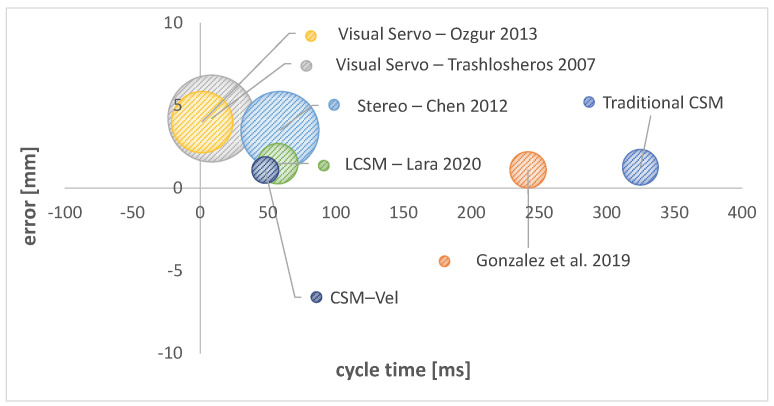
Static error vs. cycle time.

**Table 1 sensors-22-07323-t001:** Positioning error for static targets.

Error (mm)	
Average	1.086
Max.	1.36
Min.	0.76
Std. Dev.	0.195

**Table 2 sensors-22-07323-t002:** Tracking error for a target with constant speed linear motion.

Conveyor Speed	$g_{\mathrm{ii}}$	RMS Tracking Error (mm)
7 cm/s	2.3	11.2137
	2.7	9.6279
	3.1	8.8899
9.5 cm/s	2.3	14.8628
	2.7	12.3435
	3.1	11.7613
12 cm/s	2.3	27.8823
	2.7	20.9840
	3.1	18.6458

**Table 3 sensors-22-07323-t003:** Tracking and final positioning errors for freehand moving target along different trajectories.

Trajectory	RMS Tracking Error (mm)	Final Error (mm)
circle	7.521	1.44
square	10.471	1.41
decreasing spiral	9.021	1.23
lemniscate	9.661	1.16
zig-zag	10.788	1.29

**Table 4 sensors-22-07323-t004:** Cycle time and tracking error comparison.

Control Cycle Time [ms]	Static Error (mm)	Std Dev (mm)	Tracking Error (mm) @ vel (mm/s)	Method
325	1.26	0.34	NA	Traditional CSM
242	1.11	0.35	NA	Gonzalez et al. [18]
8.33	4.21	2	20@800	Visual Servo—Trashlosheros [41]
NA	0.4	0.21	NA	CSM—Bonilla [42]
1.4	4	1	NA	Visual Servo—Özgür [43]
58.8	3.5	1.6	NA	Stereo—Chen [44]
57.3	1.48	0.43	NA	LCSM—Lara et al. [23]
48	1.09	0.19	8.89@70, 11.76@95, 18.65@120	CSM—Vel

## Data Availability

Not applicable.

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
