# Peer review of "A Novel Velocity-Based Control in a Sensor Space for Parallel Manipulators"

_sensors, 2022, doi:10.3390/s22197323_

Round 1
Reviewer 1 Report (Previous Reviewer 1)
I have no further comments. It can be published in current form.
Author Response
Please see the attached file.

Reviewer 2 Report (Previous Reviewer 2)
1) In the previous submission of the manuscript the reviewer pointed out
"Velocity based control of a parallel manipulator seems to be nothing new. This idea goes back to Huynh et al in 1997 [Huynh-97]. Moreover, the concept of current paper seems to be already published in [Balmaceda-16]. The Kalmanfilter based approach for estimation of camera-space manipulation parameters is also described in [Gonzalez-19] - by some of the same authors. All references are highly relevant but missing the manuscript. Moreover, what is the novelty with respect to those - if there is any?"
The authors replied to above questions. However, their reply is not reflected in the current version of the manuscript. Above highly relevant references (Huynh-97,Balmaceda-16,Gonzalez-19) are missing in the current submiassion, as well. This has to be done.
2) Previoously, the reviewer also pointed out
"... In the present manuscript there is no single comparison of the performance (accuracy, computation time, smoothness) of the proposed parallel robot with resepct to an equivalent single robot. so what exactly is the gain of the robot, if there is any? ..."
The authors replied "The current submission shows a new velocity based CSM control strategy, it is not meant to compare the
performance of the parallel robot to any other platform." which is unsatisfactory.
It is recommended to compare and present memory consumption and computational complexity of the proposed version with respect to existing versions in accordance with the authors' comment in lin 82, "This probabilistic approach allows us to obtain satisfying results with a minimal amount of data history and computational effort." A conjecture without proof has no merit.
Above issues have to be resolved before the paper is ready for publication.
References (to be included):
Huynh-97 DOI 10.1109/IROS.1997.655150
Balmaceda-16 DOI 10.5772/61942
Gonzalez-19 DOI 10.1177/1729881419842987
Round 2
Reviewer 2 Report (Previous Reviewer 2)
the authors considered the reviewer's comments
This manuscript is a resubmission of an earlier submission. The following is a list of the peer review reports and author responses from that submission.
Round 1
Reviewer 2 Report
The paper presents a parallel robot that is controlled via camera-space manipulation. The major concerns of the reviever are novelty as well as quality of presentation.
1) Velocity based control of a parallel manipulator seems to be nothing new. This idea goes back to Huynh et al in 1997 [Huynh-97]. Moreover, the concept of current paper seems to be already published in [Balmaceda-16]. The Kalmanfilter based approach for estimation of camera-space manipulation parameters is also described in [Gonzalez-19] - by some of the same authors. All references are highly relevant but missing the manuscript. Moreover, what is the novelty with respect to those - if there is any?
2) The authors claim that their robot using two cameras controls parallel actions. Two cameras can only control one robot i.e., all arms of the parallel robot have to do exactly the same job. In the present manuscript there is no single comparison of the performance (accuracy, computation time, smoothness) of the proposed parallel robot with resepct to an equivalent single robot. so what exactly is the gain of the robot, if there is any? Fig.7 and Fig.8 only show that the proposed robot works in principal.
3) The authors claim in the abstract "Velocity-based control has several advantages compared to position-based control: it yields a smoother motion, it is more energy efficient, it is more suitable for low control frequencies and is more robust to variations on the control signal." Also this conjecture has no proof. There needs to be a discussion comparing accuracy in mm, computation time in ms, smoothness in frames/seconds, control frequency in Hz, etc., for the proposed velocity-based control system with respect to the basic point-based control system, to support the conjecture.
Without benchmarking, the proposed work has limited merit.
References:
Huynh-97 DOI 10.1109/IROS.1997.655150
Balmaceda-16 DOI 10.5772/61942
Gonzalez-19 DOI 10.1177/1729881419842987
Reviewer 3 Report
- Please provide an overview of the state-of-the-art methods (in the past three years).
- All symbols have to be clearly defined.
- Please analyze the processing speed of the proposed scheme.
- Performance evaluation: Comparisons among the proposed scheme and the state-of-the-art methods are required.